# The Development of Rubber Tapping Machines in Intelligent Agriculture: A Review

**Hui Yang** [1] , **Zejin Sun** [1] , **Junxiao Liu** [1,2] , **Zhifu Zhang** [1] and **Xirui Zhang** [1,2,*]

[1] Mechanical and Electrical Engineering College, Hainan University, Haikou 570228, China
[2] Sanya Nanfan Research Institute, Hainan University, Sanya 572025, China
* Correspondence: zhangxr@hainanu.edu.cn

**Abstract:** In the past decade, intelligent technologies have advanced rapidly, particularly due to improvements in automatic control, which have had a significant impact on forestry, as well as animal husbandry and the future of farm management. However, the degree of production and management mechanization in natural rubber plantations is low, especially since the methods of tapping still rely heavily on labor. The decrease of skilled rubber tappers and the increase in labor costs have led to the development of the mechanization of rubber tapping operations. The application of emerging intelligent agricultural technologies could provide an alternative in order to maximize the potential productivity of natural rubber. Based on this vision, we reviewed the literature on rubber tapping from the past decade for system implementation in rubber plantations. In this review, selected references on rubber tapping were categorized into several directions of research, including rubber tapping machines, the key technologies applied in tapping operations, and some related protective research, analyzing research works from 2010 to 2022 that focused on tapping methods. The review also discusses the application of intelligent agricultural technologies, such as the recognition of tapping trajectory and tapping path planning. A summary of challenges and future trends is also provided in this study. Based on the relevant research, the use of intelligent technologies in rubber tapping machines is still in its initial stage and has broad prospects. Through this study, we aim to provide a reference for researchers in the field of rubber tapping machines and thus to play a positive role in future rubber tapping.

**Keywords:** rubber plantations; natural rubber; rubber tapping machine; automatic control; intelligent agriculture

## 1. Introduction

Natural rubber is an important industrial raw material and strategic resource, the products of which are important in the fields of transportation, national defense, and the military industry [1]. The excellent resilience, electrical insulation properties, wear resistance, airtightness, and flexibility of natural rubber make it widely used in people's daily life, in the areas of medicine, hygiene, and so on. There are two types of rubber currently used in industry [2]. One is natural rubber, produced by natural plants, and the other is synthetic rubber, synthesized by chemical processing. Driven by industrialization, the use of synthetic rubber has developed rapidly, accounting for more than half of the total production capacity in the world [3]. Synthetic rubber is superior to natural rubber in terms of some of its special properties, such as chemical resistance and grease resistance [4]. However, the high impact resistance, puncture resistance, and tear resistance of natural rubber are unmatched by synthetic rubber. This facilitates metal bonding, which makes it irreplaceable, especially for aircraft tires and truck tires used in service under complex load conditions [5]. For instance, the manufacturing of truck radial tires requires 50% natural rubber, off-the-road (OTR) tires need to be made of 90% natural rubber, and aero tires need

to be made of 100% natural rubber. According to statistics, there are more than 70,000 types of products around the world that are made from natural rubber.

As shown in Figure 1, natural rubber is obtained from the laticiferous vessels of the rubber tree via regular tapping [6]. Tapping is the process of creating a path that must lie 1.2–2.0 mm and at an appropriate angle above the cambium, which is the tissue between the wood and laticiferous vessels [7]. The thickness consumed in the vertical direction during each tapping is defined as the bark consumption, and this is required to be 1.4–2.1 mm [8]. Thus, it is a skill-oriented job [9]. During 2:00–6:00 a.m., the latex output is higher due to the expansion pressure reaching its highest point, so this operation is usually performed in the early morning [10–12]. Tapping operations may also lead to latex allergy and occupational disease [13–16]. With the combination of the development of urbanization and industrialization, as well as great skill requirements and high labor intensity of manual tapping, the problem of dwindling rubber farmers has become increasingly prominent [17].

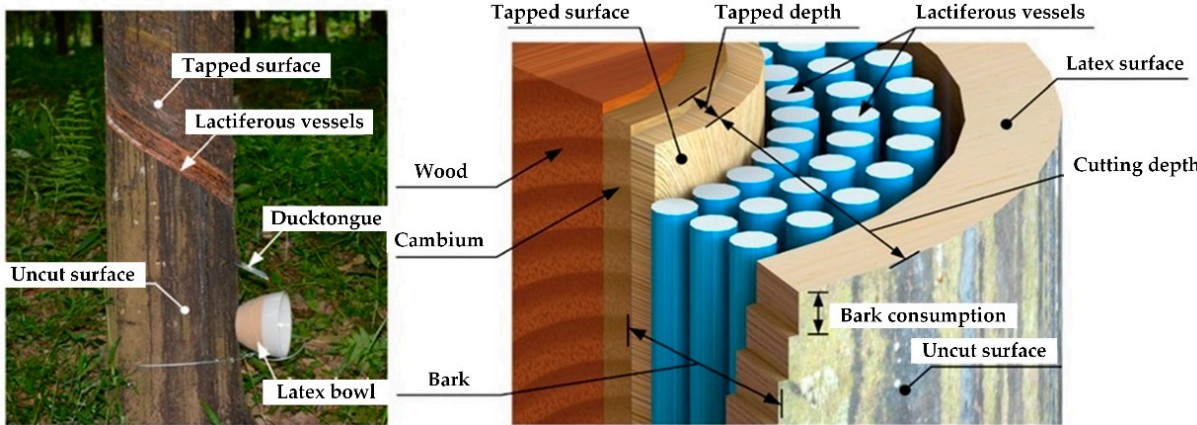

**Figure 1.** Partial views of a rubber tree trunk [18].

With the aim of responding to the above problems, emerging information technologies are important in transforming traditional farming methods into a new type of precision agriculture [19–23]. For example, CIHEVEA Technology Company Limited, Ningbo, China put forward an automatic intelligent rubber tapping system [24]. This system sends tapping instructions from a remote-controlled terminal and receives data signals through a secondary base station. Then the base station in the field transmits instructions to the machines to complete the operation of rubber tapping. This example preliminarily reveals the application of agricultural information technologies in the production management of rubber plantations. Nevertheless, a critical limitation of the system is the inconvenience of the maintenance process, due to the need to connect each rubber tree in the orchard to a machine. Thus, improving the automation and intelligence level of rubber tapping machines plays an important role in increasing the output of natural rubber [25].

Research in the area of rubber tapping has followed several avenues. Early work by McMullen [26], Ligia Regina et al. [27], and Ramachandran et al. [28] was concerned with the effect of tapping frequency and tapping environment (the living tree under completely aseptic conditions and in the absence of oxygen) on rubber yield, based on biology (symptoms of bark scaling, cracking, drying, necrotic streaking, and browning of the internal bark, leading to the decay of internal tissues). Ru et al. [29] and Wang et al. [30] found that tapping skills and tapping machinery were key factors influencing the low efficiency of labor tapping. In actual experiments, mechanical tapping can help to resolve the problems related to development. The use of an electrical tapping knife was researched at the end of the 1970s. Walaiporn et al. [31] and Chantuma et al. [32] optimized the mechanical device of the rubber tapping knife, including designs of a manual tapping knife and an electrical tapping knife, setting the depth and controlling the thickness to relieve the symptoms of carpal tunnel syndrome among rubber tappers and improve

production efficiency. The main advantages of the knives are their small volume, light weight, low power, and ease of operation. The types of electric tapping knives designed include translational cutting, rotary cutting, and intelligent manual cutting knives. With automation constraints, these knives have been suggested to be used as an auxiliary tool. In addition, works by Cheng [33] and Zhou [18] deal with the design of intelligent rubber tapping technology evaluation equipment based on automatic tapping, navigation, and remote control. The emergence of this kind of self-propelled rubber tapping robot means the beginning of the liberation of labor for rubber tappers.

The application of intelligent agriculture in rubber tapping is still in the early stages of development, whereas a considerable amount of academic research has accumulated in crop planting [34–37], orchard harvesting [38–40], and field management [41–45], suggesting complex multi-dimensional impacts of intelligent agriculture. Moreover, the number of references on rubber plantations (including harvesting and protection) is very limited at present. A comprehensive review of the relevant literature will be especially helpful in synthesizing the key research insights and unveiling major research trends in this field. We thus hope to help new researchers to grasp the current state of the art by summarizing the articles available on rubber tapping machinery research. The overall contributions of this study are set out as follows. We have (1) summarized the articles covering the design and experiments concerning rubber tapping machines from 2010 to 2022; (2) discussed the advantages and disadvantages of tapping machines based on key technical indicators; (3) expounded upon the automatic tapping technologies applied in self-propelled robots to contribute to further research in the area of rubber planting.

To acquire a better understanding of the trends related to rubber tapping machines, in this paper a brief overview of the status study and development is provided first. Then the key relevant skills in relation to these machines are introduced (especially for the operation of self-propelled rubber tapping robots), including tapping technologies, the recognition of trees and tapping lines, and the perception of obstacle information in rubber plantations. Finally, the findings of the paper are summarized, followed by the development of future prospects.

## 2. Materials and Methods

### 2.1. Search Strategy

The strategy used to search for relevant articles was important in this review. In this study, we followed the Preferred Reporting Items for Systematic Reviews and Meta-Analyses (PRISMA) guidelines [46] and referred to the systematic literature review method [47]. We designed a search string and used the following databases as our key resources: IEEE Xplore, MDPI, Web of Science, Engineering Village, Springer, and ScienceDirect (as shown in Figure 2). All searches were conducted on 28 August 2022, using the following Boolean string: "rubber tapping robot" AND "rubber tree protection" OR "tapping machine" based on the titles, abstracts, and keywords. In this study, we used various search term combinations according to the criteria or limitations of each database (Table 1). No geographical restrictions were applied to the identification process, and the search period in the databases was from 1 January 2010 to 28 August 2022. We chose appropriate articles based on the following criteria: (i) closely related to the main idea of rubber tapping machinery and protective work, (ii) not a book, and (iii) not a report. Only peer-reviewed journal articles, conference articles, and dissertations were included.

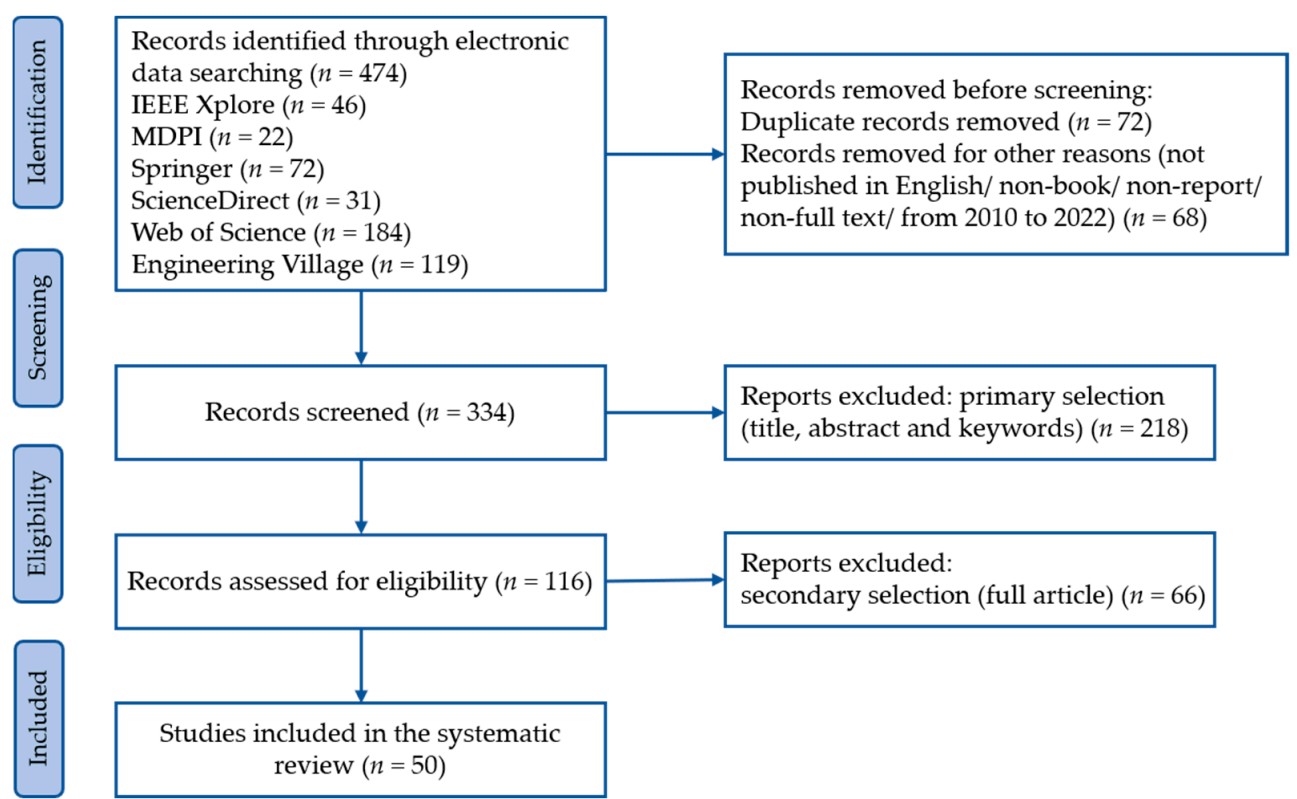

**Figure 2.** Literature selection methodology, including searching scope, selection criteria, and the number of articles included and excluded.

**Table 1.** The search strategies.

| Database | Search Terms |
|---|---|
| IEEE Xplore | Title, abstracts: "Abstract": rubber tree protection OR rubber tapping AND "Document Title": rubber tree protection OR rubber tapping. |
| MDPI | Title, keywords: rubber tapping OR rubber tree protection. |
| Web of Science | Title, abstracts: (((TS = (rubber tapping (robot OR machine))) OR TS = (rubber tree protection)) OR TI = (rubber tapping (robot OR machine))) OR TI = (rubber tree protection). |
| Engineering Village | Title, abstracts, keywords: (((rubber tapping) WN KY) OR ((tapping machine) OR (rubber tree protection) WN KY)) AND(({ja} OR {ca} OR {cp} OR {ds}) WN DT)). |
| ScienceDirect | Title, abstract or author-specified keywords: "tapping machine" OR "rubber protection". |
| Springer | Find resources: "rubber tapping" AND (machine OR technique OR robot) OR "rubber tree protection" within (Article AND Chapter and Conference Paper). |

### 2.2. Search Criteria

The full text of each paper that met the above criteria was considered in the decision regarding whether to exclude it according to its relevance to the following detailed criteria: rubber tapping operations, including tapping knives, tapping machines, tapping robots, related protection research, recognition of tapping paths, and navigation in rubber plantations. After the final selection, 50 articles were included in this review, and their findings were then cited and analyzed. We include a brief summary and categorization of the contents of the included research in Table A1 of Appendix A, including (i) protective

research to increase production in the traditional manual tapping period, (ii) the design of portable electrical tapping devices, (iii) research on fixed tapping machines, and (iv) studies on self-propelled tapping robots.

### 2.3. Data Extraction

Information about the research in the 50 articles was extracted, including articles on (i) rubber tapping machines, (ii) intelligent agricultural technologies (including the recognition of tapping paths, navigation in the orchard, and tapping methods for the protection of rubber trees), and some (iii) related tapping studies, such as the influence of the external environment on the production of natural rubber. Out of the final papers included for review, the distribution of studies published between 2011–2022 is visualized in Figure 3. The year 2018 and 2019 had the highest numbers of papers, and the number of papers showed an increasing trend year by year. According to the data compiled in Table A1, the development of rubber tapping machines and the key skills used in operations are shown in Figure 4. Due to the use of manual tapping methods before 2016, the articles from this period were based on the analysis of occupational diseases caused by long periods of wrist work and how to improve latex production from a pathological perspective. Then, a series of portable electric tapping knives and fixed tapping machines were designed in order to relieve wrist fatigue and improve the efficiency of tapping. The topic of how to navigate rubber plantations was proposed at a conference in 2010. Nevertheless, research papers on how to recognize the position of the tapping trajectory, track its preconcerted path, and finish motion planning did not appear until 2018. At the same time, the structural design of the manipulation and motion control of multi-DOF (multi-degree of freedom) systems were realized, which led to the application of self-propelled rubber tapping robots in recent years.

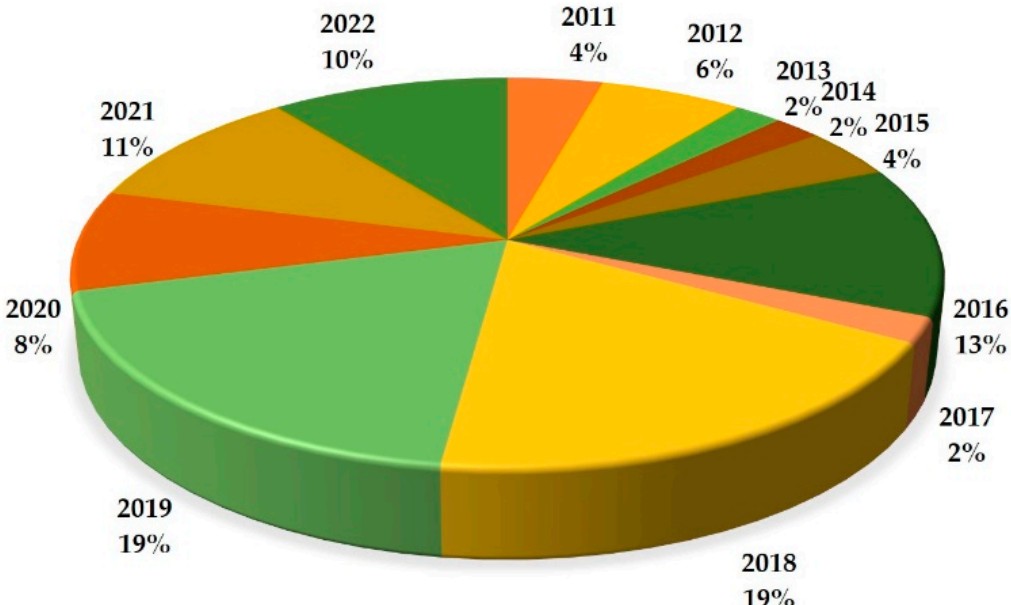

**Figure 3.** Distribution of studies in terms of their year of publication (2011–2022).

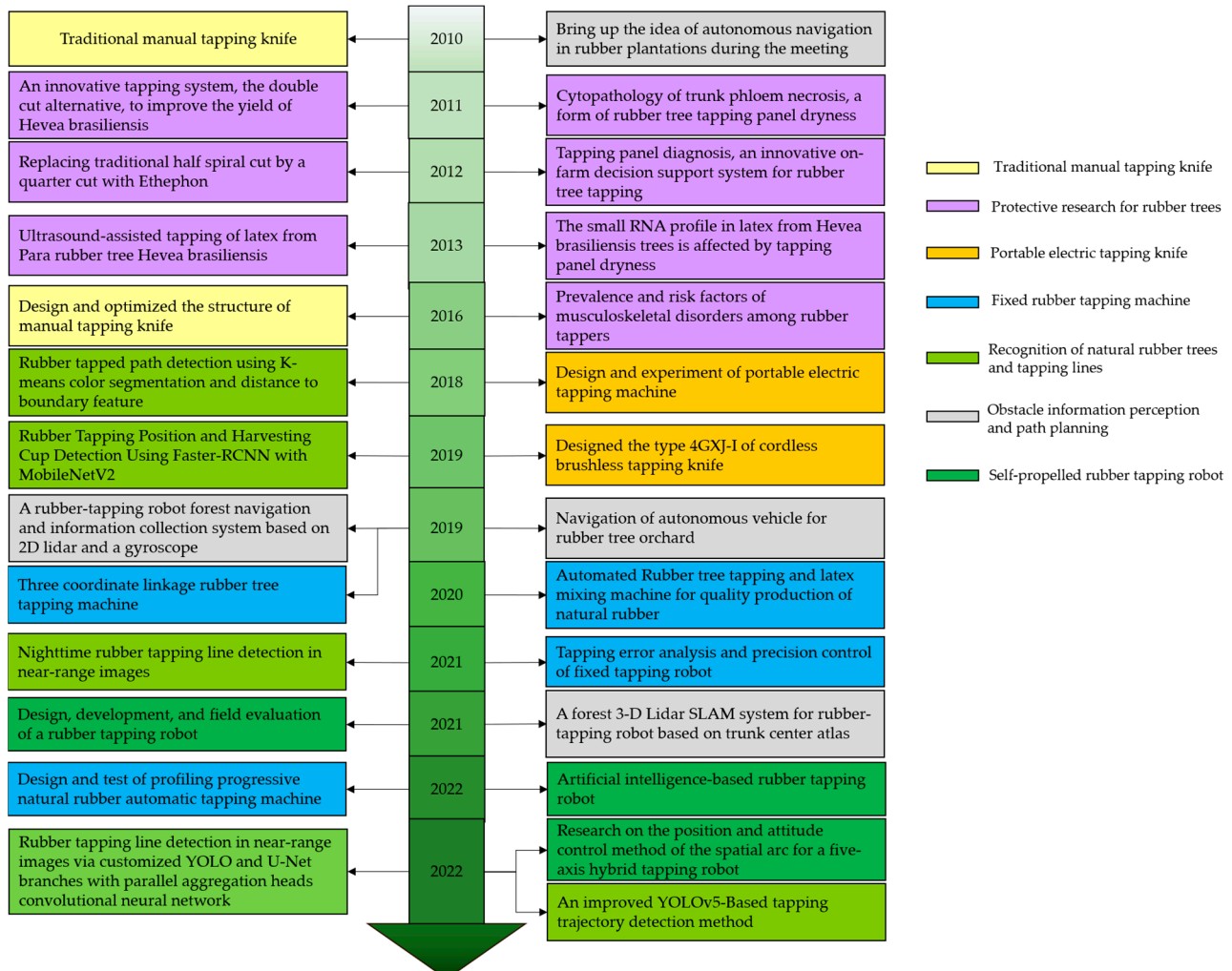

**Figure 4.** Diagram of the development of rubber tapping machines and the application of related intelligent technologies according to the research references.

## 3. Development of Rubber Tapping Machines

### 3.1. Manual Tapping Machine

According to the development stage, two methods of obtaining natural rubber appeared first. One is the method of traditional manual tapping. The other one is micro-tapping with gas stimulation, which involves the use of chemical stimulants to prolong the discharge time of laticifer on the line of tapping [48,49]. This method has not been widely applied because there are still some problems such as air leakage, bark coarsening, and secant inside shrinking. Therefore, in this section we mainly focus on the former method.

The designs of traditional tapping knives, as shown in Figure 5a, include the pull-type tapping knife and the push-type tapping knife. The use of these knives has the advantage of being low cost and ease of acceptance by workers. However, there are many shortcomings associated with this kind of tapping method. For example, the tool bit cannot be replaced. In addition, it is difficult to control the quantity and shape of a single consumption, which has seriously affected the economic benefits of rubber industries [50]. A portable rubber tapping machine is a hand-held device, which can be easily deployed in plantations. As shown in Figure 5b, the use of the 4GXJ (Rubber Research Institute, Chinese Academy of Tropical Agricultural Science) electrical tapping device, a kind of portable electrical tapping device driven by electricity, can decrease the labor intensity of rubber workers [51]. Because of its poor levels of automation, it is preferable to use it as an auxiliary device [52]. Furthermore, it does not offer a hands-free solution because it still relies on manual handling.

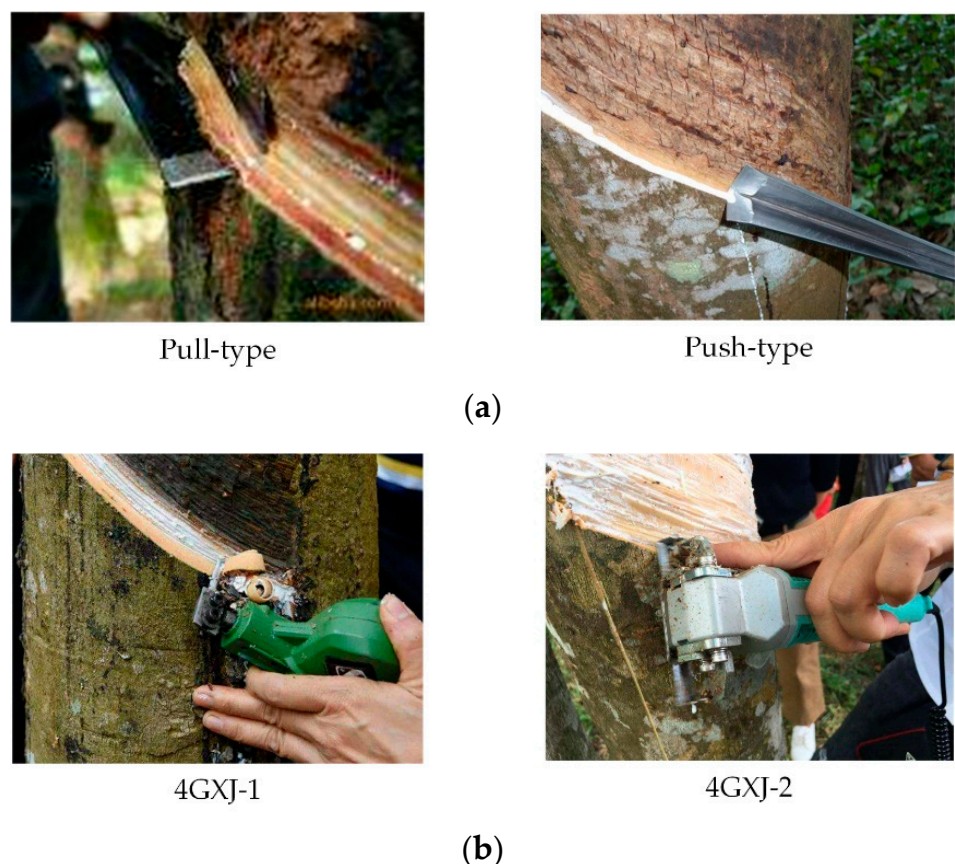

Pull-type

Push-type

(**a**)

4GXJ-1

4GXJ-2

(**b**)

**Figure 5.** (**a**) Traditional tapping knife; (**b**) portable electrical tapping device.

### 3.2. Fixed Tapping Machine

Manual rubber tapping is a time-consuming and skill-oriented type of labor. A rubber tapping machine does not completely rely on manual work, which means that the labor intensity of the work can be reduced to a certain extent [53–55]. As shown in Figure 6a, the bundled profiled tapping machine has the advantages of low cost, a lightweight structure, and a stable tapping movement. It is attached to the tree by means of straps. Easier control of the tapping depth enables the fixed tapping machine to avoid damaging trees. This kind of equipment can expediently locate the relative position of the collection cup, which can be prepared for harvesting [56,57].

Ning et al. [58] designed a fixed composite track-type rubber tapping machine (as shown in Figure 6b). By selecting the dry rubber output as the response value, the second-order regression model of the response value and the significant parameters were obtained based on the Box–Behnken design test. The results showed that when the speed of the motor was 21 r/min and the preload force of the string was 20 N, the tapping machine could obtain the rubber yield of 6.29 mL in the first 5 min with the optimal parameter combination. As shown in Figure 6c, a fixed rubber tapping machine driven by a motor was designed with an intelligent control system [30]. When the host computer gave the tapping command to the system, the tapping knife realized the presupposed spiral trajectory motion according to the designed program in the controller. At the same time, the feed rate of the tapping knife was controlled in real time using the feedback data obtained from a distance sensor to realize the depth-controlled tapping operation.

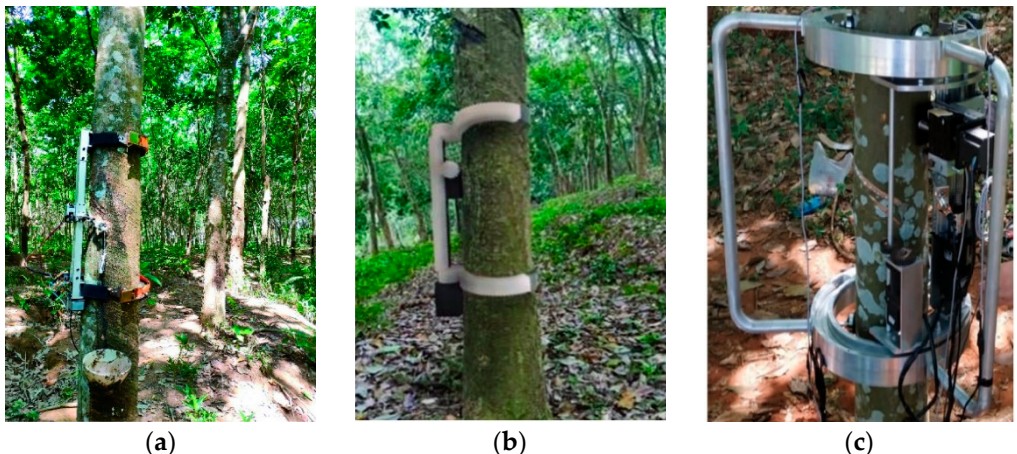

**Figure 6.** Fixed tapping machine. (**a**) Bundled profiled tapping machine [59]. (**b**) Fixed compound motion track rubber tapping machine [58]. (**c**) Rubber forest cutting test site [30].

*3.3. Self-Propelled Rubber Tapping Robot*

With the development of automation technologies, automatic control, sensor recognition, image processing, and other technologies have been widely applied in rubber tapping operations, which has promoted the design of self-propelled rubber tapping robots. As shown in Figure 7a, HARIBIT [60] unveiled a rubber tapping robot based on vision and lidar technology, which used a camera to access two-dimensional code labels attached to rubber trees. The two-dimensional code label contained the height and horizontal position information of the line to be tapped. They controlled the movement of the mechanical arm in relation to the corresponding position of rubber tapping according to the label and used a depth camera to extract the coordinates of knife marks on rubber trees to construct a new cutting track for rubber tapping.

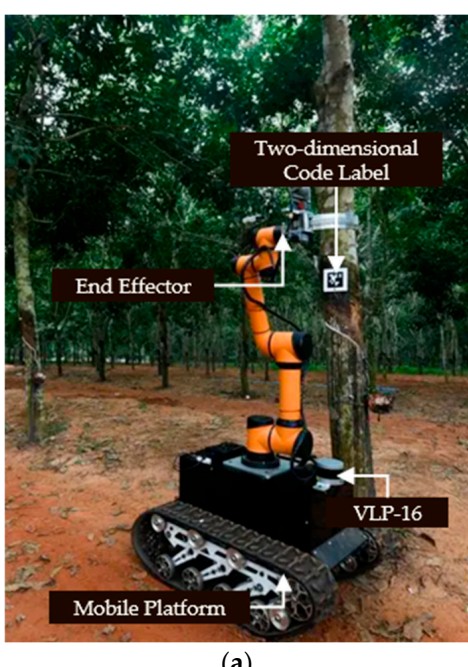 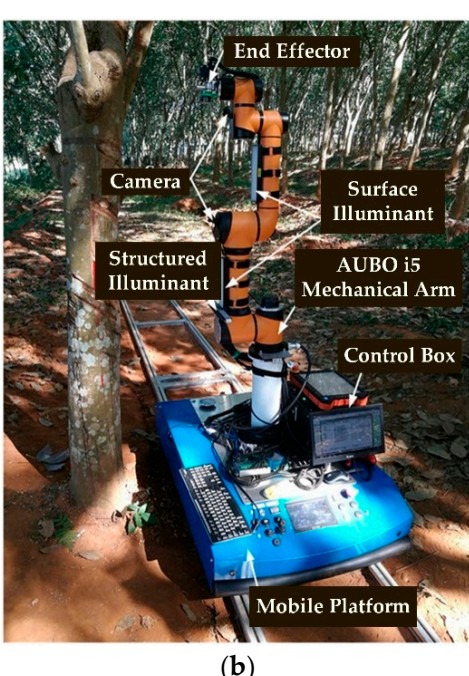

**Figure 7.** Self-propelled rubber tapping robot. (**a**) Crawler-type tapping robot [60]. (**b**) Rail-mounted tapping robot [18].

As shown in Figure 7b, Zhou et al. [18] presented a rubber tapping device with a rail-based mobile platform. This was an innovation over previous rubber tapping devices. An upgraded spatial spiral trajectory was established by analyzing the trunk and

manual rubber cutting trajectories for the operation of a six-axis tandem robot arm. A compact binocular stereo vision system was used to achieve the initial perception of the parameters required for the cutting trajectory. In addition, an integrated end-effector was developed to enable a more precise perception of the cutting trajectory and cutting operations. However, the motion of this robot depended on the rail, which greatly limited the application scenarios for the robot. Moreover, the study did not provide navigational operation information for the crawler chassis. Compared with this kind of fixed automatic tapping machine, a self-propelled rubber tapping robot is more intelligent and relies on less labor. Nevertheless, there is still room for research on these robots' navigation, control, and their recognition of rubber tapping trajectories. In the context of emerging information technology development, the use of self-propelled rubber tapping robots will become a trend in the future.

Notably, an accurate tapping track, tapping depth, tapping thickness, and profiling mechanism are the most important evaluation indexes for rubber tapping machines. We have classified the above references and summed up the indexes used in each study in Table 2. The ticks and crosses in the table indicate whether the device in the literature contains the function in the list. In terms of the current development status of agricultural robots, most of the equipment is still in the laboratory stage and has not reached the stage of industrialization and commercial use. Against this background, the research and development of agricultural robots can make breakthroughs in the following aspects. One is to develop fixed automatic rubber tapping machines. This kind of research and development does not need to move the chassis, and it is easy to carry out production tests. The other is to focus on the research and development of the operating chassis and special mechanical arms with independent positioning and control systems. An operating chassis that can plan the operating route and avoid obstacles stably would provide the premise for mobile robots to transition from the laboratory to the market.

**Table 2.** Classification of rubber cutters and summaries of evaluation indexes used in each study.

| References | Classification | Accurate Tapping Track | Tapping Depth | Bark Consumption | Profiling Mechanism | Year of Publication |
|---|---|---|---|---|---|---|
| [50] | Traditional cutting | ✖ | ✖ | ✖ | ✖ | 2011 |
| [61] | Optimized rubber tapping knife | ✖ | ✔ | ✔ | ✖ | 2018 |
| [51] | Electrical rubber tapping knife | ✖ | ✔ | ✔ | ✖ | 2018 |
| [56] | Fixed rubber tapping knife | ✔ | ✔ | ✔ | ✖ | 2020 |
| [18] | Self-propelled rubber tapping robot | ✔ | ✔ | ✔ | ✔ | 2021 |
| [59] | Fixed rubber tapping machine | ✖ | ✔ | ✔ | ✔ | 2022 |
| [62] | Self-propelled rubber tapping robot | ✔ | ✔ | ✔ | ✔ | 2022 |

## 4. Rubber Tapping Technology

Rubber harvesting stress can lead to tapping panel dryness (TPD) and can influence the non-structural carbohydrate (NSC) storage in trunk wood, which affects the production of latex [63–65]. A tapping machine needs to pay attention to the following important indicators:

(1) An accurate tapping track: because of the spiral shape used, it is necessary that the tapping tool can move according to the specified precise track. Moreover, to realize

automatic rubber tapping and save cost, the mechanical parts for rubber tapping should be simplified.

(2)  A controllable tapping depth: excessively deep rubber tapping will cause damage to the tree body, and excessively shallow tapping will affect the yield of rubber, so it is necessary to design a device that can ensure that a uniform detection and limit depth are maintained each time rubber tapping is carried out.

(3)  Reasonable bark consumption: too much consumption of bark will shorten the total tapping cycle, and too little will reduce the latex yield. Therefore, on the premise of ensuring the lactation tube, the amount of skin consumed in a single tap should be reduced as far as possible.

(4)  A stable profiling mechanism: as a rubber tree is not an ideal ellipse, it is necessary to design a copying device to make the laticifer partition more even.

### 4.1. Manual Tapping

Susanto et al. [61] optimized the use of a rubber knife to control tapping depth and used software to select component materials and conduct manufacturing tests on nine rubber trees. The treatment was carried out according to the design plan, keeping the depth between 1 and 1.5 mm of the cambium, the tapping thickness between 1.5 and 2 mm, and tilting angles between 35° and 60°. The results of the functional testing showed that rubber tapping can function properly and using intervals of 09:00–10:00 WIB for slopes 60°, 45°, and 35°, they obtained average latex amounts of 1.83 g, 1.11 g, and 0.74 g. The rubber tapping capacity of 5–6 s per tree was an improvement upon conventional rubber tapping, which required between 6–8 s with 5.32 cm$^3$ rubber bark consumption. Their blades were connected by means of bolt joints so that they could be easily replaced if a blunt force was encountered.

Zhang et al. [51] designed a portable electric tapping machine. The experimental results showed that in addition to individual data, the difference in bark consumption at each cut was less than 0.2 mm, and the average bark consumption of each group was also less than 0.2 mm. This showed that the bark consumption was uniform and could meet the requirements of the design. Furthermore, the knife body had no extrusion on the tapping line, which was conducive to rubber discharge. The design's symmetrical blade structure can be used to tap rubber with an upside and a downside, greatly reducing the technical difficulty and labor intensity of rubber workers, and changing the rubber tapping operation from a professional task to a more accessible one. Chen et al. [66] designed a series of electric rubber tapping knives called 4GXJ. By adjusting the relative position of the guide and the circular edge of the blade, precise control of the tapping depth was realized at a millimeter level. Furthermore, by moving the guide up and down and adjusting its relative position with the horizontal edge, precise control of the millimeter thickness of bark consumption was realized. The results showed that under the conditions of torques of 0.2 N·m, 0.22 N·m, and 0.25 N·m, the movement curve of the transmission mechanism changed smoothly and periodically, and the maximum stress generated by the key components was 5.147 × 10$^3$ MPa, which was much smaller than the yield strength of the material, 2.206 × 10$^8$ MPa, and there was no obvious change in deformation. A performance comparison between a traditional tapping knife and an electrical tapping knife is shown in Table 3. Because of the need for a motor, a lithium battery, transmission components, and precision requirements, the cost of an electrical rubber cutter is higher than that of a traditional one. With the same operation time or labor intensity, the tapping area can be increased by 20–30%. At the same time, the use of an electrical cutter can reduce bark consumption and the damage to trees, which can make up for the higher cost.

**Table 3.** Performance comparison between the traditional tapping knife and the electrical tapping knife [67].

| Parameter Type | Traditional Tapping Knife | | Electrical Tapping Knife | |
|---|---|---|---|---|
| | Push-Type | Pull-Type | 4GXJ-1 | 4GXJ-2 |
| Power | Manpower | Manpower | Brushless motor | Brushless motor |
| Cutting time per tree (s) | 12–18 | 12–18 | 10–16 | 5–10 |
| Bark consumption (mm/year) | 110–150 | 110–150 | 110–130 | 110–130 |
| Battery capacity (mAh)/Endurance (h) | - | - | 2000/1.5–2.0 | 4000/3.5–4.5 |
| Cost of training time for rubber tappers (d) | 25–30 | 25–30 | 3–5 | 3–5 |

*4.2. Semi-Automatic Tapping with Fixed Machine*

Deepthi et al. [56] proposed the structure of a semi-automated rubber tapping machine, the depth of which was initially set by manual movement and achieved semi-circular rotation via the motion of the wiper motor carrying a blade along with it. Driven by the motor, the small circular gear began to rotate, driving the tapping blade to move along with the large circular gear. The model was held by a circular clamp that could be attached to the tree and the structure could be adjusted with bolts and nuts according to the diameter of the tree. The model could be easily fixed to the tree and could tap a fixed spiral trajectory. The system is simple yet cost-efficient, with a one-time investment. It reduces labor costs and assessments which are prone to human error. Zhang et al. [68] designed and tested a three-coordinate linkage natural rubber tapping device based on laser ranging. The artificial tapping rubber surface was taken as the reference tapping line, and the depth information of the tapping tool was measured at several set collection points through the use of a laser ranging sensor. Test results showed that the error in terms of bark consumption was about 5%.

However, because the surface of a rubber trunk is not ideally cylindrical, its uneven surface defects increase the difficulty of rubber tapping. To overcome the problem of an uneven bark surface, a fixed automatic tapping machine was designed with a profiling mechanism [59]. In this system, driven by a decelerating stepper motor, the screw nut drives the end-effector to move in a straight line along the slider bracket, and the elliptical motion is combined with the linear motion to realize the three-dimensional spiral movement of the rubber tapping actuator along the tree trunk. A cylinder at the back end of the depth-limiting roller of the model rolls relative to the top surface of the rubber tree to copy the tapping track of the rubber tree. The use of this copying mechanism can ensure a stable spiral angle. According to Figure 8a, combined with the test data, when the motor speed was 21 r/min and the spiral angle was 25°, the output of natural rubber was 6.29 mL in the first five minutes, and this was the best parameter combination. Ning et al. [58] designed a fixed composite track-type rubber tapping machine. As shown in Figure 8b, response surface diagrams showed a bell-shaped curve with a downward opening, which indicated that with an increase in the factor level, the yield of rubber first increased and then decreased. Tests showed that when the cutting angle was 28°, the cutting depth was 5.5 mm, and the skin thickness was 1.6 mm, the dry glue output reached the best value of 178.4 g. The credibility of the analysis was also verified.

*4.3. Automatic Tapping with Self-Propelled Robot*

Zhou et al. [18] proposed a rubber tapping robot for natural rubber plantations and developed an integrated end-effector to further accurately perceive the tapping trajectory and tapping operations. The adaptability of the upgraded trajectory and the accuracy, as well as the efficiency of the starting point positioning algorithm, were verified by field experiments on a rubber plantation. Alignment error and execution time curves are shown in Figure 9. Overall, the alignment error was around $1.0 \pm 0.1$ mm, and the execution time was $17.01 \pm 3.65$ s. As shown in Figure 10, the accuracy of chip thickness and chip width was about $1.73 \pm 0.28$ mm, and $5.07 \pm 0.13$ mm, respectively. Furthermore, the accuracy

of the chip weight was 1.99 ± 0.24 mm. The average operating time of the whole tapping operation was 80 ± 5 s. That study demonstrated the potential of applying industrial robotics to the field of latex harvesting.

Moreover, researchers have also presented a method of tapping with a guide-positioning-type screw track [62]. This model uses a spiral track to realize a spiral tapping track, which is advantageous to the flow of latex. This study demonstrates the potential of applying industrial robotics to the field of latex harvesting. At present, there is growing interest in research on the autonomous operation of self-propelled rubber tapping machines. Moreover, this kind of robot has a tremendously marketable value and potential [69,70]. As shown in Figure 11, the development of a self-propelled rubber tapping machine usually requires the fusion and testing of various technologies, such as navigation, sensor perception, and automatic control. The development of an autonomous navigation technique is of great importance to the automatic movement of a self-propelled rubber tapping machine, especially in night-time operations [71]. Path planning through rubber forests is necessary in order to meet the requirements of agricultural norms and to look for reasonable walking tracks on the premise of nonredundant and non-leaking operations.

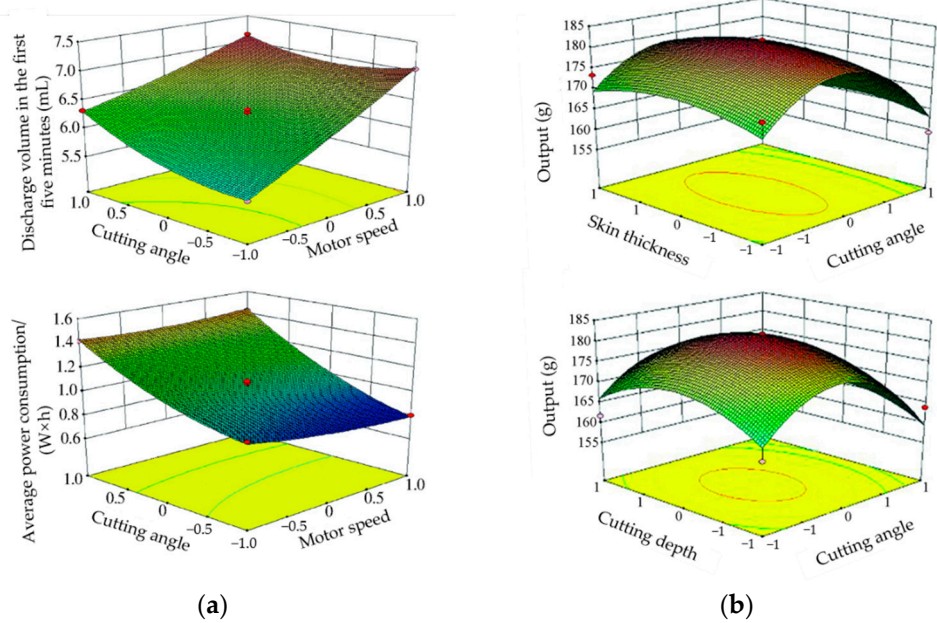

(**a**)　　　　　　　　　　(**b**)

**Figure 8.** Diagram of experiment results. The change in the color of the graph from blue to red indicates the change in the ordinate from less to more. (**a**) Interaction response surface [59]; (**b**) response surface diagram of cutting angle and cutting depth (left), and response surface diagram of cutting angle and bark consumption thickness (right) [58].

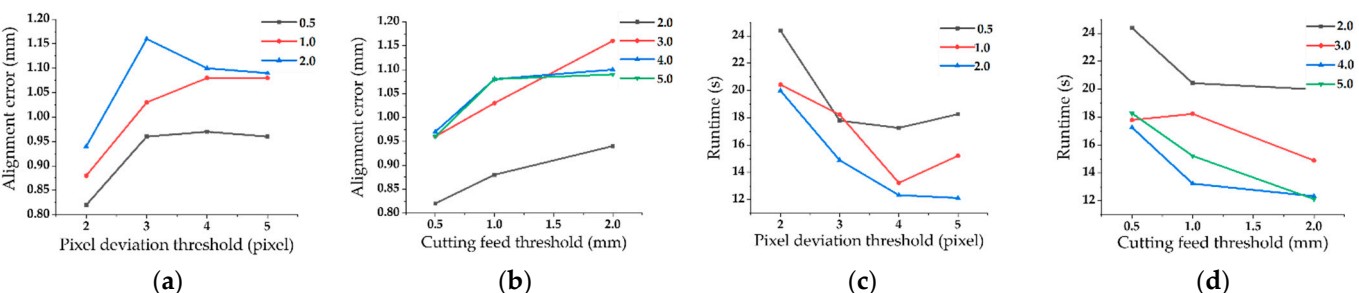

(**a**)　　　　　　(**b**)　　　　　　(**c**)　　　　　　(**d**)

**Figure 9.** Curves of multiple threshold pair test results: (**a**) pixel deviation—alignment error results, (**b**) cutting feed—alignment error results, (**c**) pixel deviation—execution results, and (**d**) cutting feed—execution results [18].

Path planning for navigation in rubber orchards commonly involves global path planning and local path planning (Table 4). Local path planning requires real-time feedback. It is used for local obstacle avoidance in path planning [72] and local tracking path planning [62,73,74] is based on sensing the external environment through sensors and feeding information back to the controller for real-time path control decisions [75]. Global path planning is a planning method that requires the complete path information of a rubber forest as prior data [76–78]. In the following section, the identification of rubber trees and rubber cutting lines and the application of obstacle information detection in path planning are described.

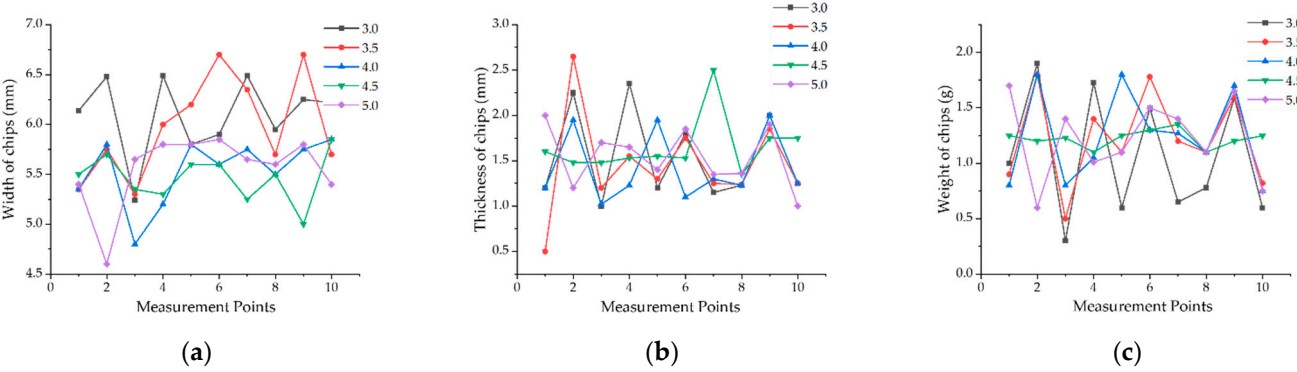

**Figure 10.** Measuring results of the cutting depth test in groups [18]. (**a**) Chip width measurement results, (**b**) chip thickness measurement results, (**c**) chip weight measurement results.

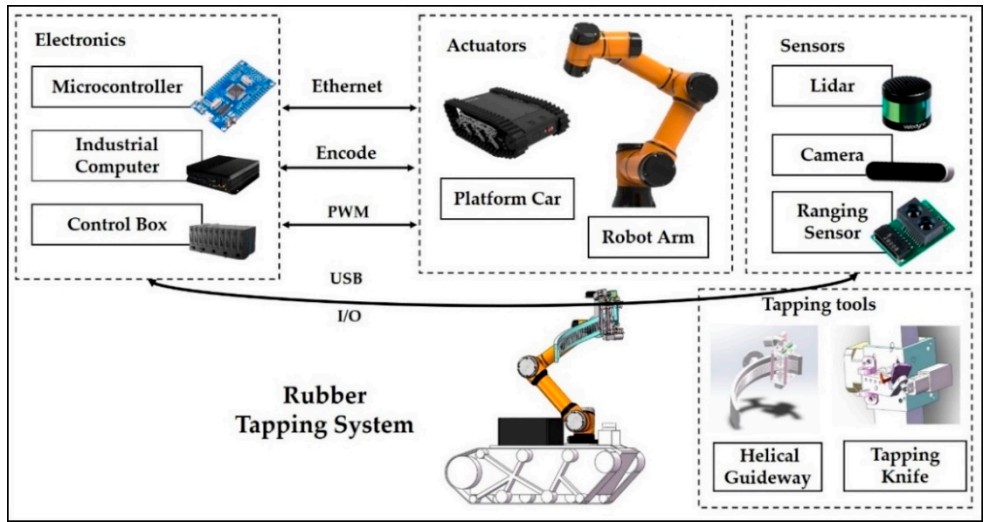

**Figure 11.** The structure of a rubber tapping robot.

4.3.1. Obstacle Information Perception and Path Planning

Obstacle information perception technology refers to the detection and identification of surrounding obstacles in an unstructured operation environment, which provides security and a guarantee of intelligent operation in a complex environment [79–82]. To date, many technologies, such as sensors, machine vision, Lidar, and ultrasonic geomagnetic position [83,84], have been developed to study the autonomous navigation of robots [85]. Some researchers have aimed to develop an algorithm of a vision system for the replacement of the expensive sensors used in typical autonomous vehicles [86–88]. The vision system model was implemented using only one camera installed on the vehicle to search for calibrated targets that were put on the trunks of rubber trees.

**Table 4.** Key components of articles related to path planning.

| References | Sensors | Feedback | Path Planning Classification | Year of Publication |
|---|---|---|---|---|
| [72] | Camera, GPS, Odometric sensors | Visual | Local tracking path planning | 2010 |
| [76] | Single camera | Visual map | Local obstacle avoidance path planning | 2018 |
| [74] | Camera, Odometric sensors | Visual, Auto-steering | Follow-the-past path tracking planning | 2019 |
| [73] | Lidar, gyroscope | The point cloud map | Local tracking, global point-to-point path planning | 2019 |
| [62] | 3-D Lidar | Point cloud | Local tracking path planning | 2021 |

Many automated systems require the use of cameras for their acquisition devices, with a common working principle of applying a vision algorithm to find a combination of image formats or point clouds [71]. Nissimov et al. [89] proposed a method to detect obstacles using a Kinect 3D sensor, which can be used for the autonomous navigation of robots to classify pixels of suspicious obstacles through color and texture features. Skoczen et al. [90] designed an obstacle detection system for agricultural mobile robots using an RGB-D camera. Al-kaff et al. [91] came up with a method to simulate human behavior and use a monocular camera to detect the collision states of approaching obstacles. Zhang and Li [92] studied mobile robot recognition and an autonomous obstacle avoidance system based on Kinect 3D sensors.

However, machine vision is greatly affected by the working environment and lighting conditions, as well as the technologies involved in the method, such as image processing, image analysis, camera calibration, and the extraction of navigation parameters, which make it rather difficult to apply in an outdoor environment in agriculture [93,94]. Rubber-tapping techniques require this activity to be carried out at night. The 3D Lidar SLAM system aids in tool mark recognition under low light conditions. Lidar can not only provide a lot of accurate information with high frequency but can also meet the requirements of accuracy and speed. In addition, it performs well compared to its cost, providing all-weather services regardless of variations in lighting conditions.

Zhang et al. [73] used a low-cost Lidar system and a gyroscope to extract the sparse point cloud data of tree trunks by means of clustering method, realizing the autonomous navigation of an intelligent rubber-tapping platform and collecting information on a rubber forest. The following activities were carried out by applying a fuzzy controller, as shown in Figure 12a,b: walking along a row with a fixed lateral distance, stopping at a fixed point, turning from one row to another, and collecting information on plant spacing, row spacing, and trees' diameters. As shown in Figure 12c, the results of three repeated experiments showed that the root mean square (RMS) lateral errors were less than 10.32 cm, and the average stopping error was 12.08 cm, indicating that the proposed navigation method provided great path tracking performance. Zhou et al. [95] proposed a random forest-based place recognition and navigation framework. Aiming at the problem of place recognition based on 3D point clouds, two kinds of features were extracted to form multi-modal feature vectors. The random forest model was applied to the place recognition of a self-propelled robot. They introduced the odometry location relationship output into the loop discrimination process. Then, the loop detection algorithm was added to S4-SLAM, forming S4-SLAM2, to realize the global localization of the mobile robot in the map. Multiple experiments were performed in an outdoor environment to confirm the proposed method, with results demonstrating its feasibility and effectiveness.

Nie et al. [62] used a tree trunk recognition system based on the optimized density-based spatial clustering of applications with noise (DBSCAN) to ensure the accuracy and

robustness of recognition in different distance scales, in the absence of GPS in forests. The algorithm (after density field correction) could make use of trees with further distances, which had been filtered out by the original DBSCAN algorithm due to the low density of point clouds. The original algorithm could only detect trees within 8.9 m on average, whereas their improved algorithm could successfully detect trees within 23.5 m on average. To further improve the environmental adaptability and feature stability of their SLAM system, the latest research achievements in visual SLAM and semantic SLAM can be integrated into their system in the future. Zhou et al. [96] introduced a sensing approach which combined Lidar sensor data with vision sensor data in a self-supervised learning framework to robustly identify a drivable terrain surface for robots operating in forested terrain. In this system, the acquisition frequency of Lidar sensor data is low, and thus it is not only used for ground recognition but can also be used to automatically supervise the training of a classifier based on visual features (color and texture). Experiments showed that the sensor system revealed good sensing performance in several forested environments.

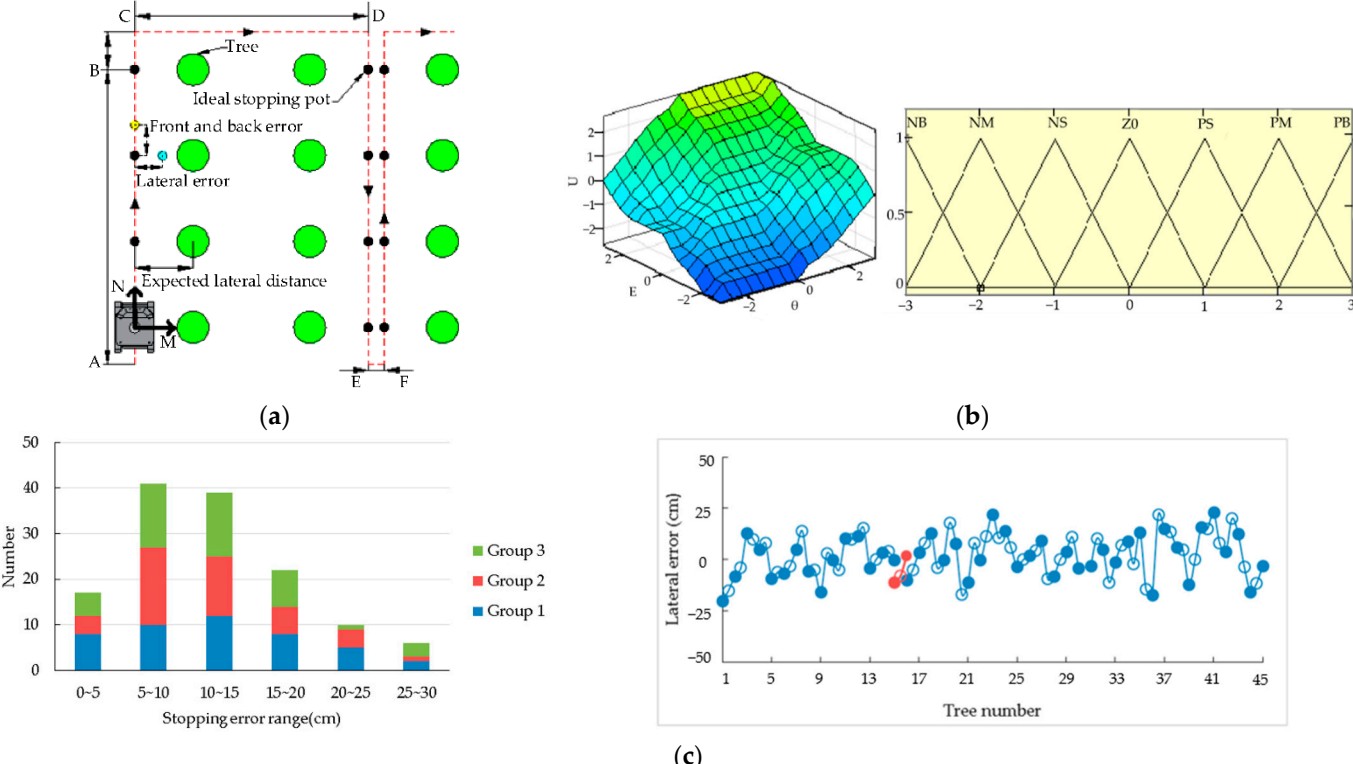

**Figure 12.** (**a**) Diagram of navigation paths used in tapping robot tasks [73]; (**b**) diagram of fuzzy control rules (left) and the distribution of the membership function (right) [73]; (**c**) distribution of stopping error and lateral error [73].

Among these method, the combination of visual range and artificial intelligence algorithms can improve the identification accuracy, but it is vulnerable to factors of the external environment, such as light interference [85]. The ability to lower the cost of a camera to increase the distance and scope of measurements is limited. Furthermore, the 3D laser radar method has high measuring accuracy and good stability, but its cost is high. As a result, information detection based on multi-sensor fusion is a possible direction of research. In addition, autonomous mobile intelligent rubber tapping robots face some technical difficulties, such as poor recognition accuracy, poor adaptive ability, navigation difficulties, and servo control with a weak light source, which need to be solved in the future.

### 4.3.2. Recognition of Natural Rubber Trees and Tapping Lines

The identification of rubber trees and tapping lines is an important aspect of local navigation in rubber plantations. Therefore, agricultural robots are required to identify a rubber tree body in the process of forest navigation, stop by the rubber tree to be cut, and find the relative position of the cutting track to plan the next path for rubber tapping. In the beginning, some researchers used image processing techniques to preprocess collected pictures of cutting parts and segment them according to simple features, such as discontinuous pixels, shapes, and characteristics of color. To ensure the objectivity and classification of the classification of the tapping panel dryness symptoms, Li et al. [97] used image processing technology to study the segmentation of tapping lines and pictures of latex. In order to eliminate the interference of other factors in the secant segmentation process, they first precut the secant image, including clipping, angle, and scale processing. On this basis, the maximum variance method was applied to segment the secant and the background, and the latex and the background, and experiment s were conducted to verify the effectiveness of this method. However, the secant and latex segmentation algorithms still need to be improved and optimized further due to the complex illumination conditions in the practical application environment.

With the development of neural network algorithms, the research direction has shifted from image processing to supervised learning or unsupervised learning, with the aim of using a network to recognize the characteristic region. The question of how to improve the accuracy and robustness of such algorithms in different scenarios has become a problem to be solved. Wongtanawijit and Kaorapapong [86] developed a simple tapped line detection algorithm using the color of wood for segmentation and the shape of the tapped area on the rubber tree as the features to be recognized by a linear SVM classifier model. Scene images were acquired by means of a light-assisted smartphone camera. The SVM classifier performance showed that the features extracted with the linear binary classifier reached 70% precision with 60% recall in some configurations. The performance was worse in some configurations as well. The identification accuracy error of this method was considerable, but this method can provide an idea for future research on the detection of trees and tapping lines. This feature can be combined with other features such as texture features to enhance the performance of the model in classifying this type of data. Some researchers [87] proposed a tapping-path and latex cup detection algorithm using a Faster-RCNN detector, which is a state-of-the-art precision detector. Their acquisition tool integrated an RGB-D camera with assisting lights for the capturing of images under low-light conditions. Fine-tuning this method for tapping-path and harvesting-cup detection, they obtained a mean average precision (mAP) of 0.95 at 0.5 intersections over union (IoU). They proposed a simple data-preprocessing step (pixel-wise multiplication) before putting the images into CNN detection networks, which meant that their detection results exhibited higher average precision. This is a well-known architecture that is lightweight in size and computation that has displayed high feasibility in regard to its deployment in actual mobile platforms.

Wongtanawijit and Kaorapapong [88] also used the sub-array searching technique to detect a tapping line consistent with the downward tapping system. The average discrete Hausdorff distance was used to measure the detected distance errors of the tapping lines. Using the intersection-over-union ratio of 0.5 as the evaluation criterion for comparing the pre-defined ground truth and the detected bounding boxes, they achieved the highest detection accuracy, up to 90.1%. These techniques feasibly support the generation of a tapping path in future studies. Worawut and Akapot [76] developed a model of a vision mapping system which is suitable for rubber tree plantations. The vision model was designed for the use of a single camera to capture calibrated targets that were placed on rubber tree trunks. The experiments showed that the percentages of error displayed small differences throughout the test range. Moreover, using a higher-resolution camera in the subsequent step could greatly improve the mapping performance.

## 5. Conclusions and Future Trends

There is tremendous potential for an increase in interest in the field of intelligent rubber tapping machines. In this paper, we first expounded on the development of rubber tapping machines in three aspects: manual rubber tapping devices, fixed semi-automatic rubber tapping machines, and self-propelled automatic rubber tapping robots. Then, the typical tapping method of each kind of tapping machine was represented, indicates that the development of automatic tapping robots with intelligent technologies at their core, is a suitable direction for the future research on tapping operations in rubber plantations. In this paper, we also introduced the applications of emerging agricultural technology in relation to automatic tapping robots, mainly including the recognition of tapping trajectories and path planning. Finally, we draw conclusions and identify the future scope of research.

According to the status of research on the independent operation of rubber tapping machines, due to the geographical factors affecting rubber growth, the current research on agricultural machinery equipment in rubber plantations is limited, and most automatic rubber cutting robots are in the laboratory stage. At present, the development of intelligent rubber tapping robots for the management of rubber plantations should be accelerated to realize the management of mechanization, as well as intelligent and automatic directional development. Moreover, the operating system of agricultural machinery equipment in rubber plantations is standardized, and functions can be called or extended according to the application background and needs of the project, which is convenient for users seeking to standardize management, and thus the work of rubber plantations can be carried out in an orderly manner. Furthermore, with the development of "3S" (remote sensing, geography information, and global positioning systems) technology, it is inevitable that computer science techniques such as learning algorithms, image processing, and so on, will be applied to the processing of data after obtaining data. In order to overcome the obstacles facing the use of sensors in the operation of agricultural machinery and equipment, multi-sensor fusion detection is a focus of future research on obstacle perception for equipment in forest operations.

In the future, the optimization of rubber cutting machine robots should not only realize more efficient and accurate rubber cutting operations, but could also consciously transform rubber cutting from a single operation to a cooperative cluster operation, from an isolated information node to an intelligent information system, and from an overall platform to a modular platform. Moreover, navigation connected to the Internet of Things (IoT) has been widely applied to unmanned ecological farm management; however, there is still a long way to go in the rubber plantation domain. The integration of the planting, plant protection, and harvesting of rubber plantations can bring about agricultural benefits and affect the future development trends of rubber plantations.

In conclusion, the use of rubber tapping machines for intelligent agriculture can not only deliver agricultural benefits but could also alleviate the problem of the shortage of rubber tapping workers. This review provides a brief reference on the research status of rubber tapping machinery and could thus play a positive role in the sustainable development of natural rubber in the future.

**Author Contributions:** Methodology, H.Y. and Z.Z.; writing—original draft preparation, H.Y. and Z.S.; writing—review and editing, H.Y. and Z.Z.; project administration, X.Z. and J.L.; funding acquisition, X.Z.; visualization, H.Y. and Z.S.; validation, H.Y. and J.L. All authors have read and agreed to the published version of the manuscript.

**Funding:** This research received financial support from Academician Innovation Center of Hainan Province, China, grant number (YSPTZX202008); Key research and development projects of Hainan Province, China, grant number (ZDYF2021XDNY198); China Agriculture Research System, grant number (CARS-33-JX2); Science research projects of Hainan Colleges and Universities, China, grant number (Hnky2022-9).

**Institutional Review Board Statement:** Not applicable.

**Informed Consent Statement:** Not applicable.

**Data Availability Statement:** Not applicable.

**Conflicts of Interest:** The authors declare no conflict of interest.

## Appendix A

**Table A1.** A summary of the 50 articles included after the final selection stage.

| Paper ID | Information Extraction and Future Work | Category | Year |
|----------|----------------------------------------|----------|------|
| S01 [72] | The topic of a machine which can autonomously navigate rubber plantations with obstacle detection capabilities was raised in the conference. | Navigation in rubber plantations | 2010 |
| S02 [50] | Based on different methods and principles, stability analyses of 25 superior rubber genotypes showed agreement, indicating stable genotypes. The study was backed up by ample data. | Protective research in traditional manual tapping period | 2011 |
| S03 [32] | An innovative tapping system, the double-cut alternative, to improve the yield of *Hevea brasiliensis*. | Innovative rubber tapping method | 2011 |
| S04 [49] | In this study, a two-stage field experiment was conducted to evaluate a wide range of low-intensity harvesting systems based on ethephon stimulation and the extension and application of this method was proposed. | Innovative rubber tapping method | 2012 |
| S05 [96] | In this paper, a self-supervised sensing approach was introduced in an attempt to robustly identify a drivable terrain surface for robots operating in forested terrain. The sensing system employed both Lidar and vision sensor data. | Tree trunk detection for navigation | 2012 |
| S06 [54] | This study showed that tapping panel diagnosis, used as a decision support tool, can increase remaining tapping years. The method formalized here will be a useful support for the innovating tapping management schemes. | Protective research in traditional manual tapping period | 2012 |
| S07 [13] | In this paper, ergonomic factors related to low back pain in rubber tappers was defined. The study aimed to evaluate the prevalence of musculoskeletal disorders. | Protective research in traditional manual tapping period | 2012 |
| S08 [55] | This work proved that the use of ultrasound technology, an innovative stimulation technique, as a preprocess on the tapping cut surface of the rubber trees could increase latex and dry rubber yields. | Innovative rubber tapping method | 2013 |
| S09 [98] | Compared the difference between tapped and untapped trees to find whether tapping operations had an influence on hevea rubber trees. | Protective research in traditional manual tapping period | 2013 |
| S10 [99] | Identifying pathogenicity genes in the rubber tree anthracnose fungus colletotrichum gloeosporioides through random insertional mutagenesis. | Protective research of plant disease control | 2013 |
| S11 [100] | Studied the regulation of MIR genes during latex harvesting and TPD. | Protective research | 2013 |
| S12 [53] | The authors of this study selected *Hevea brasiliensis* as their research object. The low-frequency tapping experiment proved that, compared with the standard tapping technique, low-frequency tapping technology could make up for the shortage of tapping labor in rubber cultivation. | Optimized latex harvesting technologies | 2014 |
| S13 [101] | Reviewed and collected the problem of stem and root-rot disease problems in rubber plantations. Obtained management strategies based on successes and failures. | Rubber-tapping-related work | 2014 |

**Table A1.** *Cont.*

| Paper ID | Information Extraction and Future Work | Category | Year |
|---|---|---|---|
| S14 [48] | This study set out to assess biochemical and histological changes, as well as changes in gene expression, in latex and phloem tissues from trees grown under various harvesting systems. The predicted function for some ethylene response factor genes suggested that some candidate genes play an important role in regulating susceptibility to TPD. | Protective research in tapping operations | 2015 |
| S15 [52] | This paper discussed a semi-automatic rubber tapping machine which was a battery-powered tool with a specially designed cutting blade and guide mechanism, supported by a sensory system which assisted the operator in performing tapping of the required quality and standards on all trees in a plantation. | Portable electrical tapping device | 2016 |
| S16 [5] | Questionnaires were administered to rubber tappers to measure musculoskeletal disorders (MSDs) and potential associated factors. The tests showed that MSDs were common among rubber tappers. | Protective research for rubber tappers | 2016 |
| S17 [71] | This paper presented a novel tree trunk detection algorithm that used the Viola and Jones detector, along with a proposed preprocessing method, combined with tree trunk detection using depth information. | Tree trunk detection for a self-propelled rubber tapping robot | 2016 |
| S18 [94] | This paper described a method of monocular visual recognition to help small vehicles navigate between narrow rows. | Navigation technology | 2016 |
| S19 [69] | This paper merged fuzzy visual serving and GPS-based planning to obtain the proper navigation behavior for a small crop-inspection robot. | Navigation technology | 2016 |
| S20 [102] | Annual growth increment and stability of rubber yields in the tapping phase in rubber tree clones. The results showed that annual girth growth occurred at the expense of rubber yields. | Protective research for rubber trees | 2016 |
| S21 [33] | This paper presented the design of an intelligent rubber tapping technology evaluation equipment based on a cloud model. | Assessment of tapping level | 2017 |
| S22 [103] | *Bacillus subtilis* B1 was shown to have potential biological control ability against various mildew, decay, and stain fungi in rubber trees. | Protective research of plant disease control | 2018 |
| S23 [51] | A portable electrical tapping device was designed in this paper. The method of image processing was used to verify the bark consumption of the electric tapping machine. | Portable electrical tapping device | 2018 |
| S24 [76] | This study aimed to develop a model of a vision mapping system which was suitable for rubber tree plantations based on a common farming platform in Thailand. | Tree trunk detection for a self-propelled rubber tapping robot | 2018 |
| S25 [86] | The authors developed a simple tapped line detection algorithm using the wood color for segmentation and the shape of the tapped area on a rubber tree as the features for recognition via a linear SVM classifier model. | Tapping path detection for a self-propelled rubber tapping robot | 2018 |
| S26 [97] | In this paper, image processing technology was used to separate the secant and latex to avoid interference factors, and obtain the exact secant and latex binary image. By calculating the area ratio of the corresponding binary images, the grade of TPD could be classified accurately. | Tree trunk detection for a self-propelled rubber tapping robot | 2018 |

| Paper ID | Information Extraction and Future Work | Category | Year |
|---|---|---|---|
| S27 [93] | By focusing on the tree canopy and sky of an orchard row, an unmanned ground vehicle was able to extract features that could be used to autonomously navigate through the center of the tree rows. The machine vision algorithm developed in this study showed the potential to guide small utility vehicles in orchards in the future. | Navigation technology for a self-propelled rubber tapping robot | 2018 |
| S28 [104] | The authors observed a correlation between DNA methylation status and rubber yield and related characteristics in *Hevea brasiliensis* tapped at different heights. They evaluated the effects of tapping-cut heights on rubber yield and related traits. | Protective research of rubber trees | 2018 |
| S29 [87] | This paper presented the detection of rubber tree (*Hevea brasiliensis*) tapping positions (tapping-paths) and trunk-mounted harvesting cups in RGB-D images, representing the machine vision part of an automatic rubber tapping system. | Tapping path detection for self-propelled tapping robot | 2019 |
| S30 [105] | Using an image segmentation methodology, the ratio of the tapping area to the latex area was calculated to analyze the degree of TPD. | Protective research of rubber trees | 2019 |
| S31 [61] | The design of a flexible rubber tapping tool with settings regalted to depth and thickness control was carried out to increase the productivity of rubber crops in the study region. The treatment was carried out as follows: controlling the depth between 1–1.5 mm of the cambium, keeping the thickness tapping at 1.5–2 mm, and using tilting angles of 35°–60°. | Fixed tapping machine | 2019 |
| S32 [68] | A three-coordinate linkage rubber tapping device was designed and tested, and a motion path planning method based on a short tapping cut was proposed. The planning process for the cutting path fused the information of the tapping cut and the cutting depth. Test results showed that the cutting depth was well controlled, with no damage to rubber trees and the error in terms of bark consumption was about 5%. | Fixed tapping machine | 2019 |
| S33 [106] | This paper introduced the progress and frontiers related to tapping technology, and analyzed and summarized the research on semi-automatic tapping machinery and automatic tapping machinery. The 4GXJ-I tapping knife, which is more suitable for industrial markets, was also designed by this team. | Portable electrical tapping device | 2019 |
| S34 [3] | This study summarized the achievements of the past two decades in understanding the biosynthesis of natural rubber. | Protective research for rubber trees | 2019 |
| S35 [73] | The authors made a robot walk along one row at a fixed lateral distance, stop at a fixed point, and turn from one row into another. They discussed a method using a low-cost two-dimensional (2D) Lidar and a gyroscope. | Navigation in rubber plantations | 2019 |
| S36 [74] | The authors investigated an autonomously guided robotic vehicle platform moving along a rubber tree orchard row. The navigation of the autonomous vehicle in a rubber tree orchard was successfully evaluated in terms of the magnitude of errors. | Navigation in rubber plantations | 2019 |
| S37 [56] | The authors proposed an automated rubber tree tapping and latex mixing machine for the high-quality production of natural rubber. During the tapping process, a tapping tool was used to make a depth of 4.0–4.5 mm. | Self-propelled rubber tapping robot | 2020 |

**Table A1.** *Cont.*

| Paper ID | Information Extraction and Future Work | Category | Year |
|---|---|---|---|
| S38 [17] | This study examined the effect of a high-frequency tapping system on latex yield, biochemistry, and tapping panel dryness (TPD). After conducting experiments in three locations, the results of latex diagnosis showed relatively unhealthy rubber trees as they were impacted by the high-frequency tapping system. The farmer should consider high-frequency tapping and ensure good decision-making in regard to tapping system applications. | Protective work | 2020 |
| S39 [107] | The authors confirmed that there is a correlation between the tapper height, the tapping postures of tappers, and the occurrence of musculoskeletal disorders among tapping workers. | Rubber tapping related work | 2021 |
| S40 [66] | Transmission structure design and motion simulation analysis of a 4GXJ-2 electric rubber cutter. Test results showed that the cutting depth was well controlled, with no damage to rubber trees and the error in terms of bark consumption was about 5%. | Portable electrical tapping device | 2021 |
| S41 [88] | This article presented a near-range machine vision technique for rubber tapping automation that detected the tapping line in near-range images. The authors conducted nighttime rubber tapping line detection in near-range images. Their acquisition tool integrated an RGB-D camera with assisting lights in order to capture images under low-light conditions. | Tapping path detection for a self-propelled tapping robot | 2021 |
| S42 [62] | The authors presented a novel 3-D Lidar SLAM system for rubber tapping robots. The system achieved the same real-time performance as state-of-the-art algorithms even without IMU information. | Navigation in rubber plantations | 2021 |
| S43 [90] | As a representative case, the autonomous mobile robot considered in this work was used to determine the working area and to detect obstacles simultaneously, which was a key feature for its efficient and safe operation. | Obstacle detection for navigation | 2021 |
| S44 [108] | The authors recognized the tapped area and untapped area using an improved YOLOv5-based tapping trajectory detection method. | Tapping path detection | 2022 |
| S45 [109] | Leaf hyperspectral reflectance was combined with machine learning algorithms to detect and classify the level of South American Leaf Blight, as well as predicted disease-induced photosynthetic changes in rubber trees. | Rubber-tapping-related work | 2022 |
| S46 [18] | The authors presented a rubber tapping robot with a six-axis tandem robot arm and a compact binocular stereo vision system. The bark consumption-cutting depth settings of 2.0 and 5.0 mm were more appropriate for the rubber tapping robot. The authors suggested that future work could include improvements in system stiffness and robustness. | Self-propelled rubber tapping robot | 2022 |
| S47 [110] | The authors designed an intelligent rubber tapping machine (RTM), and investigated whether the structural vibration level was suitable for the real tapping process. | Rubber tapping machine | 2022 |
| S48 [111] | A self-propelled rubber tapping robot was proposed that could move along a row of trees according to a predetermined path and tap each rubber tree. | Self-propelled rubber tapping robot | 2022 |

**Table A1.** *Cont.*

| Paper ID | Information Extraction and Future Work | Category | Year |
|---|---|---|---|
| S49 [112] | This paper used two-dimensional light detection and ranging (Lidar) and a ranging sensor to locate a space position. In the field tests, the lateral error of positioning was less than 8.86 mm, the height error was less than 0.72 mm, and the average harvest rate was 98.18%. | Self-propelled rubber harvesting robot | 2022 |
| S50 [37] | The existing problems and perspectives related to pesticide application sprayers and physical control equipment were summarized in this study. | Protective research of plant disease control | 2022 |

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
