# Peer review of "The Development of Rubber Tapping Machines in Intelligent Agriculture: A Review"

_applsci, doi:10.3390/app12189304_

Round 1
Reviewer 1 Report
In this work, the authors have presented a detailed survey in the field of crop and tapping of rubber trees. The studies are surveyed in the period [2010-2022]. Finally, the authors have concluded that the application of intelligent agriculture could be the alternative for maximizing the potential productivity of natural rubber. In the reviewer's opinion, the manuscript is very attractive and interesting. However, it should be improved to enhance the quality, as follows:
1) In Section 1, the authors have focus on introduced some research in the area of natural rubber tree farming and rubber tapping but forget about the technologies and techniques of smart agriculture. In the opinion of the reviewer, to highlight the topicality of this work, the author should mention these problems. These issues were presented in a recent study, the authors should cite it in: IoT-Enabled Smart Agriculture: Architecture, Applications, and Challenges, Applied Sciences, 2022.
2) All Figures should be enhanced to the resolution to a minimum of 300 dpi. They are rather difficult for readers.
3) The authors should write additionally some of the main contributions of this work in Section 1. It will help the reader more convenient follow a long survey.
4) In Figure 2, the authors have presented the surveyed articles. They claimed that IEEE Xplore (46), MDPI (44), Web of Science (64), etc. In the reviewer's opinion, this issue is not suitable because it is confusing a publisher (IEEE Xplore, MDPI, etc.) with a journal ranking unit (Web of Science). Maybe IEEE Xplore and MDPI journals are also in the WoS ???
Author Response
Dear Reviewer,
We quite appreciate your favorite consideration and insightful comments. Now we have revised our manuscript entitled “Intelligent Agriculture Enabled Automatic Operation in Rubber Plantations: A Review for Rubber Tapping Machines” (ID: applsci-1867602) exactly according to your comments, and found these comments very helpful. we hope this revision can make our paper more acceptable. The main corrections in the paper and the responses to your comments are as follows:
Point 1: In Section 1, the authors have focus on introduced some research in the area of natural rubber tree farming and rubber tapping but forget about the technologies and techniques of smart agriculture. In the opinion of the reviewer, to highlight the topicality of this work, the author should mention these problems. These issues were presented in a recent study, the authors should cite it in: IoT-Enabled Smart Agriculture: Architecture, Applications, and Challenges, Applied Sciences, 2022
Response 1: To highlight the technologies and techniques of smart agriculture, we have added the reference ([45] IoT-Enabled Smart Agriculture: Architecture, Applications, and Challenges, Applied Sciences, 2022) which the reviewer suggested in section 1. The revised sentence is “The application of intelligent agriculture used in rubber tapping is still in the early stages of development, while a considerable amount of academic research has accumulated in crop planting [34-37], orchard harvest [38-40], and field management [41-45], suggesting complex multi-dimensional impacts of intelligent agriculture.”
Point 2: All Figures should be enhanced to the resolution to a minimum of 300 dpi. They are rather difficult for readers.
Response 2: We have updated some unclear pictures. The details are as follows:
Figure 1. Partial scene of rubber trunk [18].
Figure 2. Literature selection methodology, including searching scope, selection criteria, and the number of articles.
Figure 4. This diagram of the development of rubber tapping machines and the application of related intelligent technologies according to the research references.
Figure 5. (a) Traditional tapping knife [50]; (b) Portable electrical tapping device [51].
Figure 7. Self-propelled rubber tapping robot.
Figure 8. Diagram of experiment results.
Figure 12. (a) The diagram of navigation path in the tapping robot tasks [74]; (b) The diagram of fuzzy control rules [74]; (c) The distribution of stopping error and lateral error [74]; (d) Local scene mapping results in Scene 1 [96]; (e) Local scene mapping results in Scene 2 [96]; (f) Software overview of the SLAM system (left) and experimental results of trees detection (right) [62].
Figure 13. (a) An example of a tapped rubber tree and measured distance from x-axis to patch’s boundary plot [87]; (b) An example of acquired tapping-path bounding boxes and 0.5 IoU average precision of tapping-path detection of each data group [88]; (c) This diagram shows the camera and the lights in XY plane (left), and the plot compares detection accuracies of their algorithm to other deep-CNN based detectors evaluated at various IoU (right) [89] ; (d) The percent error of X-direction and Z-direction at target length 0.7, 0.5 and 0.3 meters (left), and uncertainty of X-direction and Z-direction at target length 0.7, 0.5 and 0.3 meters (right) [77].
Point 3: The authors should write additionally some of the main contributions of this work in Section 1. It will help the reader more convenient follow a long survey.
Response 3: The main contributions of this work are added at the end of the fifth paragraph in Section 1 (lines 108 to 113). The detailed contents are as below.
“The overall contribution of the research is set out as follow: (1) Summarized the articles covering the design and experiments of rubber tapping machine from 2010 to 2022; (2) Discussed the advantages and disadvantages of tapping machines based on the key technical indicators; (3) Expounded automatic tapping technologies applied in the self-propelled robot to contribute to further research in the rubber planting area.”
Point 4: In Figure 2, the authors have presented the surveyed articles. They claimed that IEEE Xplore (46), MDPI (44), Web of Science (64), etc. In the reviewer's opinion, this issue is not suitable because it is confusing a publisher (IEEE Xplore, MDPI, etc.) with a journal ranking unit (Web of Science). Maybe IEEE Xplore and MDPI journals are also in the WoS?
Response 4: We are very sorry that we didn’t take this into account in the original manuscript. With this problem in mind, we reviewed databases for literature results and found that due to different search keywords, in the reviewed articles, IEEE, MDPI, and WoS also have non-duplicate articles. Absolutely, there are many duplicate articles in these databases, so we have removed them in the first step of the filtering process. In addition, to enrich the articles reviewed, Springer Link data has been added. The specific screening process is shown in Figure 2, and the contents are updated as follows.
“We designed a search string and used the following databases as our key resources: IEEE Xplore, MDPI, Web of Science, Engineering Village, Springer, and ScienceDirect (as shown in Figure 2). All searches were conducted on 28 August 2022, using the following Boolean string: “rubber tapping robot” AND “rubber tree protection” OR “tap-ping machine” based on the titles, abstracts, and keywords.” (lines 126 to 128)
Special thanks to you for your good comments.
We tried our best to improve the manuscript and made some changes in the manuscript. These changes will not influence the content and framework of the paper. And here we did not list the changes but used the “Track Changes” function of MS Word.
We appreciate your and the Editors’ warm work earnestly and hope that the correction will meet with approval. Once again, thank you very much for your comments and suggestions.
Kind regards,
Hui Yang

Reviewer 2 Report
The authors reported a review of rubber tapping tools, machines, and robots. They looked at different technologies from two different perspectives: 1) development of rubber tapping machines and 2) Rubber tapping technology.
The title of the paper doesn't reflect the material as it is pointing only to intelligent tapping operations. The review is claimed to be systematic and it must be mentioned in the title. I suggest following PRISMA guidelines. There is no navigation pane and line numbers in the paper which makes the review process difficult. Please ensure that you are using formal English language terms and phrases in the manuscript.
The review system design looks good but the diagram (Fig. 2) doesn't look correct. There are some technical terms that must be defined in the text to make the reading process easier for readers. Almost all figures are low resolution and the texts inside the images are not legible. I put more detailed comments in the attached file.

Author Response
Dear Reviewer,
We quite appreciate your favorite consideration and insightful comments. Now we have revised our manuscript entitled “Intelligent Agriculture Enabled Automatic Operation in Rubber Plantations: A Review for Rubber Tapping Machines” (ID: applsci-1867602) exactly according to your comments and found these comments very helpful. we hope this revision can make our paper more acceptable. The main corrections in the paper and the responses to your comments are as follows:
Point 1: There is no navigation pane and line numbers in the paper which makes the review process difficult.
Response 1: We are very sorry for the trouble caused to your review because the original manuscript did not add the line number. The revised article has added the line number for your convenience.
Point 2: The title covers just the intelligent machines but you have discussed all machines: manual, semi-manual, etc. It this is a systematic review, must be mentioned in title.
Response 2: We have changed the title to “Intelligent Agriculture Promotes the Development of Rubber Tapping Machines and Realizes Automatic Operation: A Systematic Literature Review”. This review mainly describes the development of rubber tapping machines, including manual, semi-manual, etc. The revised title is trying to describe a trend that intelligent agriculture promotes the development of tapping machines. Instead of describing the specific research content of the article, the revised title is more suitable to express the main idea.
Section 1 Introduction
Point 3: “laticiferous cell” What is it? A brief explanation in one or two sentences can help. Also, add a reference to Fig. 1 in the text.
Response 3: The word ‘’laticiferous cell “was replaced with “laticiferous vessels”. The revised sentence is “As shown in Figure 1, natural rubber is obtained from the laticiferous vessels of the rubber tree via regular tapping [6].” (lines 52 to 53).
And “laticiferous vessels” is added to Figure 1.
Point 4: “cambium layer” What is it? A brief explanation in one or two sentences can help.
2) What is it? A brief explanation in one or two sentences can help. Also, add a reference to Fig. 1 in the text.
Response 4: The word ‘’cambium layer “was replaced with “cambium”. The revised sentence is “Tapping is the process of creating a path that must lay above the cambium layer for 1.2-2.0 millimeters at an appropriate angle, which is a tissue between the wood and laticiferous vessels [7].” (lines 53 to 55).
And “cambium” is added to Figure 1.
Point 5: “bark consumption” what is it? Define it please.
Response 5: The revised sentence is “The thickness consumed in the vertical direction of each tapping is defined as the bark consumption, required as 1.4-2.1 mm [8].” (lines 55 to 56).
And “bark consumption” is added to Figure 1.
Point 6: The text in Figure 1 is blurred.
Response 6: We have updated Figure 1. Partial scene of rubber trunk [18].
Section 2 Materials and Methods
Point 7: You may follow the PRISMA protocol.
Response 7: The revised sentence is “This study followed the Preferred Reporting Items for Systematic Reviews and Me-ta-Analyses (PRISMA) [46] and referred to the systematic literature review method [47].” (lines 122 to 125).
Point 8: Figure 2 is not very intuitive. the arrows look like input data but they are not actually. Check the high-quality review literature and use a similar diagram.
Response 8: We revised Figure 2 according to PRISMA protocol in the manuscript.
Point 9: Figure 4 looks like a nice diagram but it is difficult to understand it intuitively. The texts are blurred. You may add a little more explanation to the diagram. What are the meanings of each arrow direction, etc?
Response 9: We have added a brief explanation of Figure 4.
The revised sentence is “Due to the manual tapping method before 2016, the articles were based on the analysis of occupational diseases caused by long periods of wrist work and how to improve the latex production from the pathological perspective. And then, the series of portable electric tapping knives and fixed tapping machines have been designed in order to relieve wrist fatigue and improve the efficiency of tapping. The topic of how to navigate in rubber plantations has been proposed at the conference in 2010. Nevertheless, the research papers on how to recognize the position of tapping trajectory, track its pre-concerted path, and finish motion planning have appeared until 2018. At the same time, the structural design of manipulator and motion control of Multi-DOF (multi-degree of freedom) were realized which led to the application of self-propelled rubber tapping robots in recent years.” (in the lines 164 to 174)
Figure 4 has also been updated in the manuscript.
Section 3 Development of Rubber Tapping Machines
Point 10: “4GXJ” mention manufacturer and the company location in parenthesize.
Response 10: The revised sentence is “As shown in Figure 5b, the 4GXJ (Rubber Research Institute, Chinese Academy of Tropical Agricultural Science) electrical tapping device...” (lines 193 to 194).
Point 11: “from the source forward” what does it mean? It is better to write in a more clear way.
Response 11: The revised sentence is “And it doesn’t come up with a hand-free solution because it still relies on manual hand-holding.” (lines 197 to 198).
Point 12: Figure 5 is low resolution.
Response 12: Figure 5 has been updated in the manuscript. We selected more representative pictures to replace the former one.
Point 13: Is it a robot? what is a "pace position”?
Response 13: We are very sorry to use the wrong words here. “robot” has been replaced with “machine” in line 207. And “pace position” has been replaced with “relative position” in the line of 208. The relative position between tapping machine and the fixed tapping machine is invariable, so it can expediently locate the position of the collection cup.
Point 14:” By selecting the dry rubber output...” what was the result?
Response 14: The sentence of the result was added in lines 213 to 216. The revised sentence is “The results showed that when the speed of motor was 21 r/min and the preload force of string was 20 N, the tapping machine could obtain the rubber yield of 6.29 mL in the first 5 min with the optimal parameter combination.”.
Point 15: What is intelligent control? what does it mean? Is it a technical term?
Response 15: The explanation of “an intelligent control” was added to the manuscript. The revised sentence is “When the host computer gave the tapping command to the system, the tapping knife would realize the presupposed spiral trajectory motion according to the designed program in the controller. At the same time, the feed rate of tapping knife was controlled in real-time by using the feedback data obtained from a distance sensor to realize the depth controlled tapping operation.” (lines 217 to 221).
Point 16:” fur” is it "FOR"?
Response 16: We are very sorry to use an inappropriate word here. The right mean is “the depth of cutting feed”. However, we have changed the expression. The revised sentence is “... by using the feedback data obtained from a distance sensor to realize the depth controlled tapping operation.” (lines 220 to 221).
Point 17: “The development of automation technologies, such as ...” check the sentence grammar. “So, the self-propelled rubber tapping was born.” Not an appropriate phrase I think.
Response 17: The revised sentence is “With the development of automation technologies, automatic control, sensor recognition, image processing, and other technologies have been widely applied in rubber tapping operations, which promotes the design of self-propelled rubber tapping robots.” (lines 228 to 230).
Point 18:” label information” what is it?
Response 18: “label information” is replaced to “two-dimensional code label”. And the “two-dimensional code label” was added in Figure 7(a). The revised sentence is “As shown in Figure 7a, HARIBIT [60] unveiled a rubber tapping robot based on vision and lidar, which used the camera to obtain the two-dimensional code label attached to rubber trees. The two-dimensional code label contained the height and horizontal position information of the line to be tapped.” (lines 234 to 238)
Point 19: The texts are not clear in figure 7.
Response 19: We have updated Figure 7 in our manuscript.
Point 20:” recognition” recognition of what? Please be specific.
Response 20: We are very sorry that we have not understood the problem here. The sentence is “...and recognition of rubber tapping trajectory...”. It means the recognition of rubber tapping trajectory.
Point 21: “feeler mechanism” What is it? Doesn't look like a technical/scientific term in this context.
Response 21: We are very sorry to use an unsuitable word here. “feeler mechanism” is replaced with “profiling mechanism”. It is a scientific term.
The explanation of the profiling mechanism: As the rubber tree is not an ideal ellipse, it is necessary to design a structure called the profiling mechanism, to make the movement of the knife along with the surface of the trunk.
Point 22: Please ensure that table 2 is easily readable.
Response 22: We have revised Table 2 in the manuscript.
Point 23: “bark consumption” Must be defined earlier in the manuscript.
Response 23: We have defined this word in line 56, the same as point 5.
Point 24: “feeler mechanism” Change it. You can use other words e.g. sensing capability, etc.
Response 24: We are very sorry to use an unsuitable word here. “feeler mechanism” is replaced with “profiling mechanism”. The reason is the same as Point 21. The hook or fork in Table 2 means whether the tapping machine in the first column has been designed with a profiling mechanism.
Section 4 Rubber Tapping Technology
Point 25: “Harvesting” changes to “Rubber harvesting”. A brief definition of tapping panel.
Response 25: The revised sentence is “Rubber harvesting stress can lead to tapping panel dryness (TPD) and influence...”.
Tapping panel dryness (TPD) syndrome is the latex vessel has loosed some or all ability to produce latex, which is manifested as the reduction of secant. This is a scientific term. Tapping panel is the surface that has been tapped.
Point 26: ‘WIB’ what is that?
Response 26: WIB (Waktu Indonesia Barat) is West Indonesian standard time, which is used as one of the standard times in the UTC+7 time zone.
Point 27: Low resolution. the texts are not readable in Figure 13.
Response 27: We have revised Figure 13 in our manuscript.
Section 5 Conclusions and Future Trends
Point 28:” secondary programming” Must be already defined in the context. Otherwise, don't use it.
Response 28: The revised sentence is “Moreover, the operating system of agricultural machinery equipment in rubber plantations is standardized, and functions can be called or extended according to the application background and needs of the project, which is convenient for users to standardize management, and the work of rubber plantations can be carried out in an orderly manner.” (lines 580 to 584).
Point 29:” Sooth to say” doesn't look formal here.
Response 29: “Sooth to say” is replaced with “In conclusion”.
Special thanks to you for your good comments.
We tried our best to improve the manuscript and made some changes in the manuscript. These changes will not influence the content and framework of the paper. And here we did not list the changes but used the “Track Changes” function of MS Word.
We appreciate your and the Editors’ warm work earnestly and hope that the correction will meet with approval. Once again, thank you very much for your comments and suggestions.
Kind regards,
Hui Yang

Round 2
Reviewer 1 Report
This version is outstandingly improved compared to the old version. In my opinion, it is eligible to be published.
Some minor revisions such as adding page numbers, issues, and volume, of each reference, should be performed.
Author Response
Dear Reviewer,
We quite appreciate your favorite consideration and insightful comments. Now we have revised our manuscript entitled “Intelligent Agriculture Promotes the Development of Rubber Tapping Machines: A Systematic Review” (ID: applsci-1867602) exactly according to your comments, and found these comments very helpful. The main corrections in the paper and the responses to your comments are as follows:
Point 1: Some minor revisions such as adding page numbers, issues, and volume, of each reference, should be performed.
Response 1: Thank you for your kind suggestion. We have updated the reference in the revised manuscript. The details are as follows.
- Ye, S.; Rogan, J.; Sangermano, F. Monitoring rubber plantation expansion using Landsat data time series and a Shapelet-based approach. ISPRS Journal of Photogrammetry and Remote Sensing 2018, 136, 134-143. doi: 10.1016/j.isprsjprs.2018.01.002
- Qi, D; Zhou, J.; Xie, G; Wu, Z.X. Optimizing Tapping-Tree Density of Rubber (Hevea brasiliensis) Plantations in South China. Small-scale Forestry 2016, 15, 61-72. doi: https://doi.org/10.1007/s11842-015-9308-8
- Chantuma, P.; Lacote, R.; Leconte, A.; Gohet, E. An innovative tapping system, the double cut alternative, to improve the yield of Hevea brasiliensis in Thai rubber plantations. Field Crops Research 2011, 121(3), 416-422. doi: 10.1016/j.fcr.2011.01.013
- Han, PP.; Chen, JS.; Han, Y.; Yi, L.; Zhang, YN.; Jiang, XL. Monitoring rubber plantation distribution on Hainan Island using Landsat OLI imagery. International Journal of Remote Sensing 2018, 39(8), 2189-2206. doi: 10.1080/01431161.2017.1420933
- Pramchoo, W.; Geater, A.F.; Harris-Adamson, C.; Tangtrakulwanich, B. Ergonomic rubber tapping knife relieves symptoms of carpal tunnel syndrome among rubber tappers. International Journal of Industrial Ergonomics 2018, 68, 65-72. doi: 10.1016/j.ergon.2018.06.004
- Said, M.E.; Belal, A.; Kotb, A.-E.S.; El-Shirbeny, M.A.; Gad, A.; Zahran, M.B. Smart farming for improving agricultural management. The Egyptian Journal of Remote Sensing and Space Sciences 2021, 24(3), 971-981. doi: https://doi.org/10.1016/j.ejrs.2021.08.007
- Varghese, A.; Panicker, V.V. Computer-Aided Ergonomic Analysis for Rubber Tapping Workers. In Advanced Manufacturing Systems and Innovative Product Design.; Deepak, B.B.V.L., Parhi, D.R.K., Biswal, B.B., Eds.; Publisher: Springer, Singapore, 2021; pp. 293-302. doi: https://doi.org/10.1007/978-981-15-9853-1_24
- Chong, Z.C.; Ali, W.M.A.; Mazlan, A.Z.A. Structural Vibration Study of a New Concept Intelligent Rubber Tapping Machine. In Symposium on Intelligent Manufacturing and Mechatronics.; Ali Mokhtar, M.N., Jamaludin, Z., Abdul Aziz, M.S., Maslan, M.N., Razak, J.A., Eds.; Publisher: Springer, Singapore, 2022; pp. 305-312. doi: https://doi.org/10.1007/978-981-16-8954-3_29
- Angel, T.S.; Amrithesh, K.; Krishna, K.; Ashok, S.; Vignesh, M. Artificial Intelligence-Based Rubber Tapping Robot. In Inventive Communication and Computational Technologies.; Ranganathan, G., Fernando, X., Shi, F., Eds.; Publisher: Springer, Singapore,2022; Volume 311, pp. 427-438. doi: https://doi.org/10.1007/978-981-16-5529-6_34
Special thanks to you for your good comments.
We tried our best to improve the manuscript and made some changes in the manuscript. These changes will not influence the content and framework of the paper. And here we did not list the changes but used the “Track Changes” function of MS Word.
We appreciate your and the Editors’ warm work earnestly and hope that the correction will meet with approval. Once again, thank you very much for your comments and suggestions.
Kind regards,
Hui Yang

Reviewer 2 Report
Thank you very much for revising your manuscript. The revised version reads better. However, there are a few comments that you must address. Also, please ensure that while addressing the points refer to the correct line numbers in the file you have uploaded. The line numbers that you referred to were incorrect. Please add a navigation pane and make sure that all your figures have a reasonable resolution. The text boxes in Fig. 7 are not well-organized. A part of the text is missing in some of the boxes. Fig. 4 looks better but it is not intuitive yet. Please explain the left and right boxes. Why some of the boxes are on the left and some on the right? There are more comments in the attached file.
Good luck!

Author Response
Dear Reviewer,
We quite appreciate your favorite consideration and insightful comments. Now we have revised our manuscript entitled “Intelligent Agriculture Promotes the Development of Rubber Tapping Machines: A Systematic Review” (ID: applsci-1867602) exactly according to your comments and found these comments very helpful. We hope this revision can make our paper more acceptable. The main corrections in the paper and the responses to your comments are as follows:
Point 1: Please add a navigation pane.
Response 1: The navigation pane has been added to our submitted version of Word and PDF. The navigation pane is on the left side of the file (with navigation enabled).
Point 2: Figure 2 looks much better. Please align the boxes at the left with right ones.
Response 2: Thanks for your kind advice. Figure 2 has been revised in the manuscript followed the suggestion.
Point 3: Fig. 4 looks better but it is not intuitive yet. Please explain the left and right boxes. Why some of the boxes are on the left and some on the right? Change the color of the big arrow.
Response 3: We are sorry that there is no different with the left box and the right one. Just putting them all on one side will make the picture look too long. Figure 4 illustrates some major research and articles that have happened in the last 10 years. The arrow in the middle are from 2010 to 2022, and the small arrows at the left and right around them indicate the year in which the research was published. According to the content of the research, they were classified by different colored boxes. The color of the big arrow has been changed in the revised manuscript.
Point 4: Make sure that Figure 7 has a reasonable resolution. The text boxes in Fig. 7 are not well-organized. A part of the text is missing in some of the boxes.
Response 4: We improved the resolution of Figure 7. Also, redraw the boxes in the pictures.
Point 5: Figure 8 is low resolution. Please change it.
Response 5: Figure 8 has been changed in the revised manuscript.
Point 6: Figure 9 and Figure 10 are low resolution. Please change them.
Response 6: We used the data in the reference to redraw the images by Origin Tools, due to the original pictures are not clear in the origin reference. The resolution of revised image is 1383×1065. Figure 9 and Figure 10 are revised in the manuscript.
Point 7: Please make sure that the resolution is good on Figure 12 and Figure 13. Also put the notation letters (a, b, c ...) close to the related figure.
Response 7: Figure 12 and Figure 13 are all replaced by the original images uploaded from the website of the literature. It is possible that the definition of images is reduced because it is scaled smaller. The images in the folder named picture are the original images with higher resolution. As well as the converted version of PDF is less legible than the original version of Word/ Microsoft office.
Special thanks to you for your good comments.
We tried our best to improve the manuscript and made some changes in the manuscript. These changes will not influence the content and framework of the paper. And here we did not list the changes but used the “Track Changes” function of MS Word.
We appreciate your and the Editors’ warm work earnestly and hope that the correction will meet with approval. Once again, thank you very much for your comments and suggestions.
Kind regards,
Hui Yang
